# Feasibility of working with a wholesale supplier to co-design and test acceptability of an intervention to promote smaller portions: an uncontrolled before-and-after study in British Fish & Chip shops

Louis Goffe,[1,2,3] Frances Hillier-Brown,[3,4] Natalie Hildred,[3] Matthew Worsnop,[5,6] Jean Adams,[7] Vera Araujo-Soares,[1,3] Linda Penn,[1,3] Wendy Wrieden,[1,2,3] Carolyn D Summerbell,[3,4] Amelia A Lake,[3,8] Martin White,[1,7] Ashley J Adamson[1,2,3]

For numbered affiliations see end of article.

**Correspondence to**
Dr Louis Goffe;
louis.goffe@ncl.ac.uk

## ABSTRACT

**Objectives** To explore the feasibility of working with a wholesale supplier to co-design and deliver, and to assess the acceptability of, an intervention to promote smaller portions in Fish & Chip shops.

**Design** Uncontrolled before-and-after study.

**Setting** Fish & Chip shops in northern England, 2016.

**Participants** Owners (n=11), a manager and customers (n=46) of Fish & Chip shops; and intervention deliverers (n=3).

**Intervention** Supplier-led, three-hour engagement event with shop owners and managers, highlighting the problem of excessive portion sizes and potential ways to reduce portion sizes; provision of box packaging to serve smaller portions; promotional posters and business incentives.

**Data collection** In-store observations and sales data collected at baseline and postintervention. Exit survey with customers. Semistructured interviews with owners/managers and intervention deliverers postintervention.

**Results** Twelve Fish & Chip shops were recruited. Observational data were collected from eight shops: at baseline, six shops did not promote the availability of smaller portion meals; at follow-up, all eight did and five displayed the promotional poster. Seven out of 12 shops provided sales data and all reported increased sales of smaller portion meals postintervention. Of 46 customers surveyed: 28% were unaware of the availability of smaller portion meals; 20% had bought smaller portion meals; and 46% of those who had not bought these meals were interested to try them in the future. Interviews revealed: owners/managers found the intervention acceptable but wanted a clearer definition of a smaller portion meal; the supplier valued the experience of intervention co-production and saw the intervention as being compatible with their responsibility to drive innovation.

**Conclusions** The co-design of the intervention with a supplier was feasible. The partnership facilitated the delivery of an intervention that was acceptable to owners and customers. Sales of smaller meal packaging suggest

### Strengths and limitations of this study

► This is the first study we are aware of to evaluate the feasibility of working with a wholesale supplier to co-design and deliver a public health intervention targeting hot food takeaways.

► It is also the first study we are aware of to detail the potential role that wholesale suppliers can play in improving the healthfulness of food offerings from hot food takeaways, exemplified by Fish & Chip shops.

► A mixed-method approach was employed, which successfully captured impacts of the interventions on all stakeholders.

► Data available on customer behaviours were limited and did not include takeaway food consumption.

► We had a small sample size, focused on one takeaway cuisine type and therefore the results may not be generalisable beyond the setting of Fish & Chip shops.

that promotion of such meals is viable and may be sustainable.

## INTRODUCTION

Takeaway and fast food meals, particularly from independent businesses have been found to deliver excessive energy by means of large portion sizes,[1] driving high-energy consumption,[2] which is a major public health concern.[3] Our previous work found that adults and children who ate takeaway meals at least weekly consumed 63–87 kcal and 55–168 kcal per day, respectively, more than those who consumed such meals rarely.[4] High takeaway meal consumption has been linked to weight gain[5] and diet-related diseases.[6]

In the UK, 'Fish & Chips' are culturally embedded,[7] with an estimated 10 500 shops nationally.[8] Typical Fish & Chip shop meals consist of white fish in batter and chipped potatoes, both deep fried.[9] One survey found that the median energy content of 64 Fish & Chip meals was 1658 kcal,[1] representing 79% of a woman's and 64% of a man's estimated average daily energy requirement.[10] This suggests that reducing portion sizes could be a means to promote population health.[11–14]

As most outlets use a limited number of wholesale suppliers,[15] these have substantial influence on what food is offered by independent hot food takeaways.[16] While working with suppliers provides an opportunity for intervention,[17] to date, limited research has been done.[16]

The aim of this study was to explore the feasibility of working with a wholesale supplier to co-design and deliver an intervention to promote smaller portion meals in Fish & Chip shops in northern England; and the acceptability of this intervention to shop owners/managers and their customers. It was not a process or outcome evaluation study.

## METHODS
### Intervention co-design

We approached Henry Colbeck Limited (HC), an independent specialist Fish & Chip shops wholesaler, supplying over 2500 shops across northern England and Scotland,[18] within a partnership supplying over 6000 shops across the UK.[8] We asked HC if they would work with us to co-design and lead delivery of an intervention to encourage Fish & Chip shops to improve portion control and promote smaller portion meals. Members of the research team and HC staff set out their respective positions and terms of partnership that included: for HC—responsibility for intervention development and intervention delivery (including costs), as well as data sharing; for the research team—responsibility for study design and coordination of data collection, independence of analysis and right to publish findings.

We discussed findings of our previous studies on independent takeaways[4 19–23] with HC, and in turn, they shared their knowledge, detailing meal packaging options currently used: boxes, trays and paper wrapping. It was established that large portion size meals existed across the sector, driven primarily by high competition and a desire to offer customers 'value-for-money'. We agreed to the dual-focus of an intervention to facilitate and promote: better portion control, supported through the use of box packaging that standardises portion size (in particular in comparison with paper wrapping); and active promotion of smaller portion meals. We agreed that implementation of the intervention should not incur direct costs to the participating Fish & Chip shops and that HC should promote to owners/managers primarily on the potential financial benefits of portion control and smaller portion sizes. HC recruited two owners with established smaller portion meal promotion to support intervention delivery

to detail their practical experience and financial benefits. The intervention was theorised in detail by the research team (online supplementary file A).

### Intervention description

The research team supported HC to develop a three-hour engagement event held in April 2016 at a hotel in North-East England. Fish & Chip shop owners, managers and their staff were invited to attend by HC. The programme included sessions delivered by HC and two Fish & Chip shop owners, followed by a question and discussion session.

Owners/managers were encouraged to place a greater emphasis on portion control by using box packaging and to actively promote smaller portion meals. The potential financial benefits of attracting a wider customer base and reducing portion sizes without pro-rata reductions in the price charged were stressed throughout, such as an increase in trade and higher meal profit margin. Participants were presented with a range (by size and material) of smaller portion boxes, but the choice of packaging selected was made by the owner/manager. An enhanced action-planning activity developed by the research team included a goal-setting form (online supplementary file B).[24] This included a 'public pledge', where Fish & Chip shop owners/managers detailed what changes they would make, how and when these would be delivered and how confident they were to deliver them. Owners/managers were encouraged to keep these pledges. We completed a template for intervention description and replication (TIDieR) checklist[25] (online supplementary file C).

Following the engagement event, owners/managers were offered two copies of one of two different A0 size posters promoting smaller portion meals for their shops (figure 1). These were delivered to shops within 16 days. HC suggested that one poster could be displayed in-store, and the other made visible to passers-by. Additional incentives offered by HC were 100 units of the box packaging chosen by the owner/manager and HC customer loyalty scheme points.

Additional intervention delivery was undertaken by HC sales staff who visited owners/managers who had expressed an interest in the engagement event but had not attended. An overview of the information presented at the event was provided to owners/managers and they were asked to complete the goal-setting form, offered the incentives and posters, and the recording of sales data was explained (see below).

### Recruitment to the intervention

HC purposively selected shops in northern England to be invited to the engagement event with the aim of recruiting shops: located in a range of socioeconomic settings, both within and outside major conurbations, and either known to be likely to engage or whose likelihood to engage was unknown. HC sent a postal invitation one month prior to the engagement event. This included a tailored message to the owner/manager with an invitation for them to

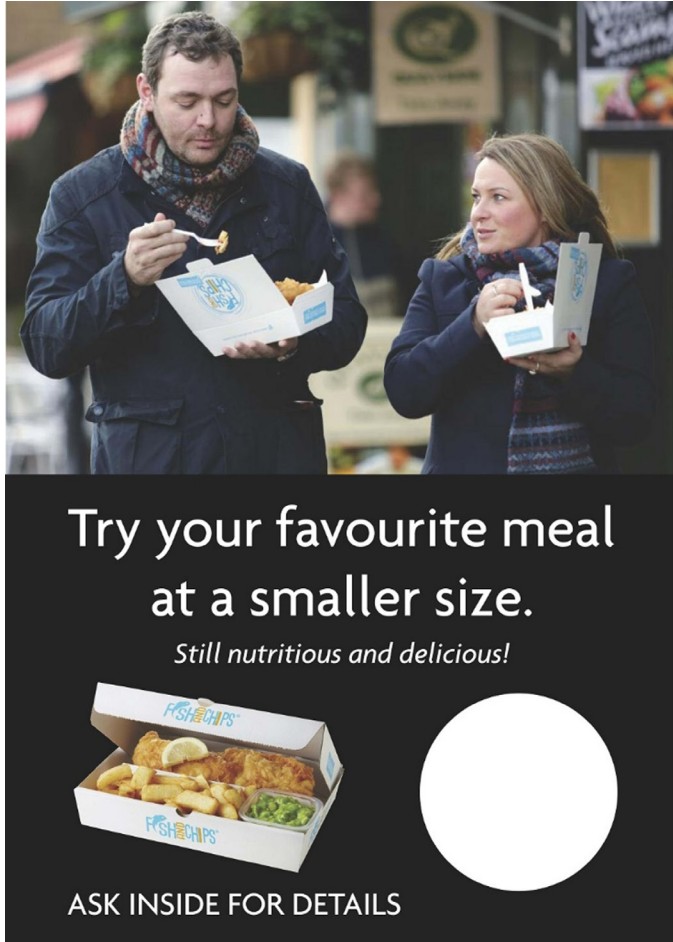

**Figure 1** Promotional A0 size poster options.

attend along with a member of their staff. Owners or managers who did not respond were contacted through phone by HC.

### Study design

We used an uncontrolled before-and-after study design to explore the feasibility of working with HC to co-design and deliver the intervention and its acceptability to Fish & Chip shop owners/managers and their customers.

### Data collection

#### Fish & Chip shop owners/managers

Data collected about owners/managers who attended the engagement event or took part in one-to-one visits included: owner or manager status; number of Fish & Chip shops owned (if owner); and whether they attended the engagement event with a member of their staff. We recorded details of each shop's: geographical location; local authority; location description (eg, city centre, village high street); Indices of Multiple Deprivation (IMD) decile of location[26]; seating provision; and Food Standards Agency (FSA) food hygiene rating.[27]

#### Goal setting

A digital image was taken of all completed goal-setting forms to record pledged changes.

#### Covert observations

To provide objective measures of change, covert data collection was completed in a subset of participating shops that were accessible to the research team, at three time points (baseline, two and six weeks postintervention). Members of the research team posed as customers. Data collected included the visibility (including the display of promotional posters) and availability of smaller portion meals. One regular size meal and one smaller portion meal (where available) was purchased from each shop. The shop menu description of the smaller portion meal was recorded as was the packaging used and meal cost. The components (battered fish and chips) of meals were weighed within two hours of purchasing.

#### Sales of Fish & Chip meals

Participating shops were provided with a booklet to record sales of regular and smaller portion meals from at least five days before to at least five days after first receiving the posters.

#### Customer survey

Following the completion of the six-week postintervention covert data collection, a customer survey was conducted in participating shops (online supplementary file D).

Interviews were conducted in-person after customers had ordered their food but before the food had been served. Questions covered customer gender, age group, awareness of the availability of smaller portion meals, views on meal portion size and purchasing behaviour including frequency of and reasons for purchasing, whether they had tried smaller portion meal and willingness to try smaller portion meals in the future.

### Semistructured interviews

All shop owners/managers who engaged in the intervention were invited to participate in a semistructured interview, either in-person or by telephone, to explore their experience of the intervention (online supplementary file D), conducted by LG. Interviews were also conducted with those responsible for the development and delivery of the intervention at HC to explore their experiences of the intervention (online supplementary file D). Interviews with HC were conducted by a researcher, FHB, not involved in intervention development and delivery.

### Data analysis

Descriptive statistics (sample size, means and proportions), conducted in R (LG), were used to summarise quantitative data but not for statistical inference.[28] Interviews were transcribed verbatim checked for accuracy and then anonymised. Thematic framework analysis with constant comparison was used to identify themes related to the feasibility and acceptability of the intervention.[29] The coding framework for each set of interviews was based on a priori themes from the interview topic guides and emergent themes from the data. The final coding framework was then applied to all transcripts, and the resulting themes were reviewed and agreed on by team members involved in the analysis (LG, AJA and MWh).

## RESULTS

### Recruitment and retention

Thirty-one Fish & Chip shop owners or managers were invited to attend the engagement event. Of these, 15 (48%) stated an intention to attend, nine (29%) attended and three did not attend but took part in one-to-one sessions with HC staff. Thus, 12 took part in the evaluation (39%) (figure 2).

### Shop setting

Shops were spread across nine local authorities. Six of the 12 shops provided seating. Shops were located in areas across all IMD deciles.[26] The FSA hygiene rating of the shops was high, with ten receiving a maximum rating of five (table 1).

### Goal setting

Eleven owners/managers completed the goal-setting form. All considered that they were already providing all of their customers with opportunities to purchase smaller portion meals in some form. The principal change to usual business practices that could be inferred from the

forms was a public pledge to promote smaller portion meals, primarily through displaying posters (n=5; table 1).

### Covert observations

We collected observational data from eight shops (table 2). At baseline, only two shops clearly displayed the availability of smaller portion meals. During at least one of the postintervention visits (two or six weeks), all eight shops displayed the availability of smaller portion meals. At baseline, one shop had smaller portion meals on their main menu, two provided smaller portion meals on their lunchtime menu only, two on their children's menu only, and two had no smaller portion meals on any menu. Postintervention, five of the eight shops actively promoted smaller portion meals using an in-store facing poster and two also displayed a poster facing outside. Of the other three: one actively promoted their own branded smaller portion meals throughout but did not display the HC posters; another had introduced a smaller portion meal by the six-week follow-up; and one only had a smaller portion on the children's menu. All but one shop used box packaging at baseline and all did so at follow-up.

Between baseline and six-week follow-up, we observed a 24 g increase in mean weight of battered fish, a 61 g decrease in mean weight of chips and a 37 g decrease in mean total meal weight of regular meals. With regards to the smaller portion meals, we observed a 2 g decrease in mean weight of battered fish, a 26 g decrease in mean weight of chips and a 27 g decrease in mean total meal weight.

### Sales of Fish & Chip meals

Seven shops returned usable sales data covering a mean of seven days predelivery and 32 days postdelivery of the posters. However, this was inconsistent in format and detailed analyses were not possible. The mean proportion of all meals sold which were a smaller portion was 14.2% preintervention and 21.2% postintervention. One shop did not return sales data due to illness, the remaining three did not provide a reason.

### Customer survey

Five owners/managers permitted customer surveys to be conducted in their shops (table 1). A total of 46 questionnaires were completed (table 3). Most customers surveyed bought meals once a month or more, choosing the shop for taste or convenience. Most were aware that smaller portion meals were available (72%) though only 20% had purchased them. Of those who had not previously tried smaller portion meals, 46% said they would be interested in trying them in the future.

### Semistructured interviews
#### Interviews with owners/managers

Interviews were conducted with eight owners and one manager, five in-person and four by telephone (table 1). Thematic analysis identified six main themes.

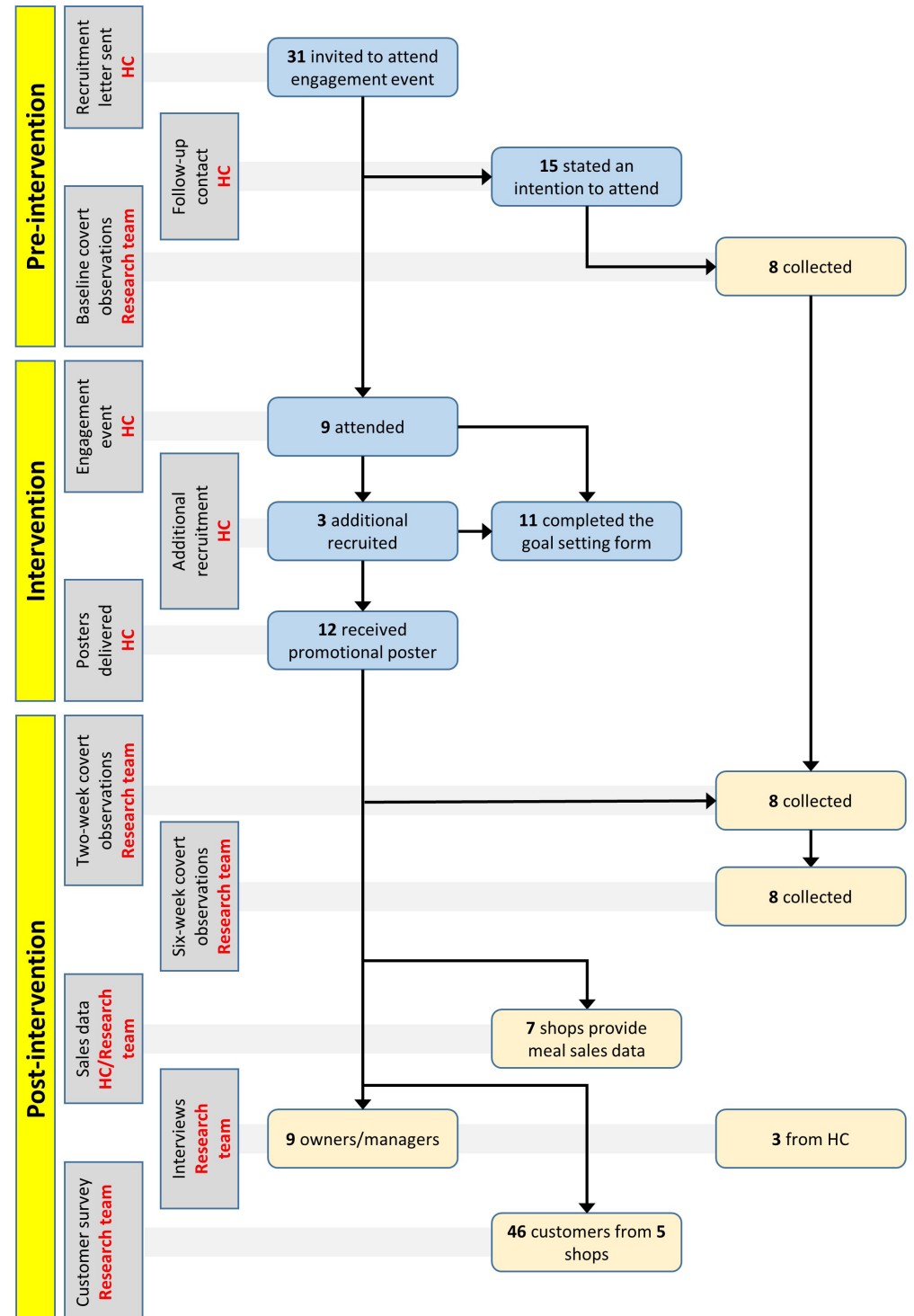

**Figure 2** Portion control intervention participation and data collection flowchart. HC,

### Relationship between owner/manager and Henry Colbeck Limited

Those who had attended the engagement event were more enthusiastic about the process than those who received the intervention in a one-to-one session. They reported that the event provided a *'unique'* [ID06, ID04] opportunity to speak about an industry matter with their peers. They were impressed with the speakers but did not value the goal-setting form. Participants felt well supported by

HC throughout. While they appreciated the incentives offered by HC they did not feel these were necessary.

### Suggested changes and smaller portion meal definitions

All respondents considered that they were already providing smaller portion meals in some form at baseline. For most, therefore, the intervention constituted the posters, whereas others reported developing a detailed

**Table 1** Summary of data collected

| | Shop-level summary (count) | Individual Fish & Chip shop | | | | | | | | | | | |
| --- | --- | --- | --- | --- | --- | --- | --- | --- | --- | --- | --- | --- | --- |
| | | ID01 | ID02 | ID03 | ID04 | ID05 | ID06 | ID07 | ID08 | ID09 | ID10 | ID11 | ID12 |
| Participant position | Owner=11 Manager=1 | Manager | Owner | Owner | Owner | Owner | Owner | Owner | Owner | Owner | Owner | Owner | Owner |
| Number of total Fish & Chip shops owned | Own 1=10 Own>1 = 2 | 1 | 1 | 7 | 1 | 1 | 1 | 1 | 1 | 1 | 2 | 1 | 1 |
| Attended the engagement event (number attendees) | Yes=9 No=3 | Yes (2) | Yes (2) | Yes (1) | Yes (2) | No | Yes (1) | No | Yes (1) | Yes (2) | No | Yes (1) | Yes (1) |
| Completed goal-setting form | Yes=11 No=1 | Yes | Yes | Yes | Yes | Yes | Yes | Yes | Yes | Yes | No | Yes | Yes |
| Public pledge | | 'Smaller box' | 'Smaller box/Display poster' | 'Smaller portion' | 'Already doing' | 'Advertising' | 'Will promote smaller portion' | 'Already using' | 'Will display promotion posters' | 'Bio box'* | Not completed | 'Already doing' | 'Display poster and use on social media to promote lite bite' |
| Covert observations conducted | Yes=8 No=4 | Yes | No | No | Yes | Yes | Yes | Yes | Yes | Yes | Yes | No | No |
| Semistructured interview | Yes=9 No=3 | Yes | Yes | No | Yes | Yes | Yes | No | No | Yes | Yes | Yes | Yes |
| Number of customers surveyed | Shop n=5 Customers n=46 | 7 | 0 | 0 | 4 | 0 | 13 | 0 | 0 | 10 | 12 | 0 | 0 |
| Semistructured interview | Yes=9 No=3 | Yes (person) | Yes (telephone) | No | Yes (person) | Yes (person) | Yes (person) | No | No | Yes (telephone) | Yes (person) | Yes (telephone) | Yes (telephone) |
| Shop region | North-East=8 Yorkshire=3 North-West=1 | North-East | North-East | Yorkshire and The Humber | North-East | North-East | North-East | North-East | North-East | Yorkshire and The Humber | North-East | North-West | Yorkshire and The Humber |
| Shop location description | | Centre of a rural village | Residential area of a market town | Residential area of a city | Shopping centre in a residential area of a metropolitan borough | Shopping centre in a residential area of a metropolitan borough | High street in a seaside town in a metropolitan borough | High street in a seaside village in a metropolitan borough | High street in a seaside town in a metropolitan borough | Centre of a rural village | High street in a seaside town in a metropolitan borough | City centre | High street in a residential area of a metropolitan borough |
| Sit-in restaurant | Yes=6 No=6 | No | No | Yes | No | No | Yes | No | Yes | Yes | Yes | Yes | No |
| IMD decile (where one is most deprived 10% of LSOAs) | | 9 | 4 | 7 | 1 | 3 | 8 | 5 | 8 | 9 | 7 | 3 | 5 |
| Shop FSA hygiene rating (0 to 5, where 0 is low and 5 is high) | | 5 | 5 | 3 | 5 | 5 | 5 | 5 | 5 | 5 | 4 | 5 | 5 |

*Bio boxes constructed from biodegradable material extracted from sugar cane.
FSA, Food Standards Agency; IMD, Indices of Multiple Deprivation.

**Table 2** Summary of covert observation data collected from each shop

| | Baseline | Postintervention | |
| --- | --- | --- | --- |
| | | 2 weeks | 6 weeks |
| Clearly displaying smaller portion meals available to all customers | Yes=2 No=6 | Yes=7 No=1 | Yes=6 No=2 |
| Smaller portion meals available to buy | Yes=6 No=2 | Yes=6 No=2 | Yes=8 No=0 |
| Active promotion of smaller portion meals | NA | Yes=5 No=3 | Yes=4 No=4 |
| Packaging used for regular meal | Boxes=7 Paper=1 | Boxes=8 | Boxes=8 |
| Weight of regular meal (g): battered fish | Mean=265.1 | Mean=277.9 | Mean=289.3 |
| Weight of regular meal (g): chips | Mean=399.9 | Mean=384.9 | Mean=339.1 |
| Weight of regular meal (g): total | Mean=665.0 | Mean=662.8 | Mean=628.4 |
| Packaging used for smaller portion meal | Boxes=6 | Boxes=6 | Boxes=8 |
| Weight of smaller portion meal (g): battered fish | Mean=175.7 | Mean=170.7 | Mean=174.0 |
| Weight of smaller portion meal (g): chips | Mean=273.0 | Mean=233.7 | Mean=247.4 |
| Weight of smaller portion meal (g): total | Mean=448.7 | Mean=404.3 | Mean=421.4 |
| Cost of regular meal (£) | Mean=£5.80 | Mean=£5.79 | Mean=£5.79 |
| Cost of smaller portion meal (£) | Mean=£4.22 | Mean=£4.07 | Mean=£4.00 |

strategy to promote smaller portion meals. One owner suggested industry-wide standards regarding portion sizes but acknowledged that *'universal adoption'* [ID12] was unlikely.

### Posters and shop setting
There was a mixed response to the posters. Some respondents felt that they were *'really good… it really just says it all… it is [poster] very relevant to our business'* [ID05], whereas others felt they did not fit with their shop's ethos. The only manager who attended the engagement event stated that the shop's owner felt the poster clashed with the shop's branding and did not display it [ID01]. Another owner was highly critical stating that *'posters that big look stupid'* and suggested a preference for alternative promotional material (eg, leaflet), detailing the *'benefits of buying smaller'* [ID06].

### Importance of quality customer service
All respondents stressed the importance of high-quality customer service in delivering smaller portion meals. One owner who had been involved in the Fish & Chip industry *'Pretty much all my life'* [ID09], had implemented numerous changes following the intervention with support from an owner who spoke at the engagement event. One owner who had been unable to attend the event reported that they had not implemented smaller portion meals in the evening as they could not rely on serving staff to deliver these consistently [ID05].

### Customer feedback
Few respondents reported receiving feedback on smaller portion meals from customers. However, one detailed the

enthusiasm from a group of builders who had seen the posters and welcomed the change [ID09].

### The ease of recording sales data
Those who provided sales data reported that this had been straight forward. However, till systems limited the value of these for analyses.

### Interviews with those who developed and delivered the intervention
Interviews were conducted with three people from HC. Thematic analysis identified five themes. In order to retain anonymity, the quotes below are not specifically attributed to a given intervention deliverer.

### Intervention deliverers' motivations regarding intervention delivery
HC representatives viewed individual shops as focused on daily sales meaning it was up to HC to take *'strategic long-term views of the industry'*. Respondents wanted shops to succeed in an increasingly competitive marketplace. They took responsibility for driving sector innovation and acknowledged that *'I need Fish & Chip shops to stay in business because they pay my wages'*.

### Considerations in the development of the intervention
HC representatives stated that effective engagement on smaller portion sizes with owners/managers could only be achieved by highlighting the financial and customer service benefits. HC staff saw themselves as providing information and choice to their customers, *"It is up to the customer [business owner] to make that choice, we are not going to force them to do anything"*. They were also happy to take

**Table 3** Customer survey responses

| Variable | Level | n (%) |
|---|---|---|
| Gender | Female | 21 (46) |
| | Male | 25 (54) |
| Age category, years | 18–30 | 10 (22) |
| | 31–40 | 11 (24) |
| | 41–50 | 2 (4) |
| | 51–60 | 8 (17) |
| | 61–70 | 7 (15) |
| | >70 | 8 (17) |
| Regular customer | Yes | 35 (76) |
| | No | 11 (24) |
| How regular | More than once a week | 2 (4) |
| | Once a week | 10 (22) |
| | Once every 2 weeks | 4 (9) |
| | Once a month | 10 (22) |
| | Once every 3 months | 6 (13) |
| | Once every 6 months | 5 (11) |
| | Once a year | 0 (0) |
| | First time | 9 (20) |
| Reasons for buying (up to two choices) | Taste/Quality | 29 |
| | Convenience | 32 |
| | Price | 3 |
| | Portion size | 0 |
| | Other | 1 |
| Portion sizes | Too small | 0 (0) |
| | Just right | 36 (78) |
| | Too big | 2 (4) |
| | NA (first time customer) | 8 (17) |
| Know about small portion | Yes | 33 (72) |
| | No | 13 (28) |
| Notice posters (where known to have been displayed) | Yes | 10 (37) |
| | No | 17 (63) |
| Tried the promoted smaller portion meals | Yes | 9 (20) |
| | No | 37 (80) |
| Try in the future | Yes | 17 (37) |
| | No | 20 (43) |
| | NA (previously tried) | 9 (20) |

the lead on intervention development and delivery and the cost to HC was viewed as an *'investment'*.

### Intervention deliverers' views on acceptability

Representatives of HC were disappointed with attendance at the engagement event (nine of 31) and were frustrated that some shops *'didn't realise the potential'*. While many had not provided a reason for non-attendance,

some reportedly told HC that it was due to staffing issues. However, the responses HC received from those who did attend were positive, *'I had quite a lot of people ring up and thanking me for the event'*, and they had viewed it as a rare *'interactive'* event: *'Fish fryers in the same room sharing ideas and you could see people writing down notes and bringing up their own problems. Other people were listening, engaging and offering advice and help. That does not happen often enough'*.

HC staff were aware that not all businesses would display the posters, with some owners/managers reporting that they were too big. However, they felt that it was important to provide the means to clearly distinguish between regular and smaller portion meals.

HC also viewed the incentive component of the intervention as an act of *'goodwill'* showing their commitment to the intervention. They did not see incentives as imperative to owner/manager involvement and, indeed, not all shops took advantage of them.

### Future plans

As a direct result of this work, HC staff developed specific packaging for smaller portion meals and associated promotional material. At the time of interviews, they were also trying to source smaller fish fillets for this new packaging.

While HC staff saw smaller portion packaging as a sustainable change, more sustainable methods of delivering the portion control message were required. The engagement event and one-to-one visits were not considered scalable or efficient, *'I can't go around and visit thousands of Fish & Chip shops because I am only one person'*.

### Experience in working with the research team

HC staff found working with the research team a positive and *'enjoyable experience'*. It had *'re-stimulated our [HC's] efforts'* and was felt to be rewarding for both parties.

## DISCUSSION
### Statement of principal findings

We found it was feasible to co-design and deliver an intervention to promote smaller portions with a commercial partner and the intervention was acceptable to both Fish & Chip shops and their customers. Attendees at the engagement event valued and enjoyed it. Shop owners/managers were broadly willing to introduce and promote smaller portion meals. We successfully measured portion sizes and collected some sales data. We observed a reduction in the size of both regular and smaller portion meals after the intervention and an increase in the proportion of meals sold that were a smaller portion. The reduction in the portion size of regular meals was due to the reduction in chips—the least nutrient-dense component of the meal. As all participating owners/managers considered that their businesses provided smaller portion meals in some form at baseline, the additional overt promotion was broadly acceptable. Most also used box packaging at

baseline meaning introduction of this was unlikely to be a key component of the intervention. The evaluation was conducted independently, avoiding the potential for competing interests of the commercial partners.

### Strengths and limitations of the study

To our knowledge, this is the first study to evaluate the feasibility of working with a wholesale supplier to co-design and deliver a public health intervention, and to demonstrate the potential role of wholesale suppliers in improving the food offerings from hot food takeaways.

Covert observations, while feasible, would be resource intensive in a larger study. While participating shops had high FSA hygiene ratings,[27] they covered a wide range of IMD deciles,[26] suggesting that the intervention may be feasible across a range of socio-economic settings. Acceptability to shops with lower hygiene ratings is unknown. Participants did not feel goal setting through the 'public pledge' was useful. Greater clarity concerning what was expected of shop owners/managers may have improved this. Some shops did not make use of posters and other marketing materials, such as leaflets, may have engaged a wider range of shops. Due to the practical constraints of the study, the customer survey had to be brief. Therefore, we did not conduct in-depth interviews with customers and these would provide more insight into their choices and preferences. Some aspects of the intervention were not felt to be sustainable by HC and further thought is required to determine how any such intervention could be scaled-up. We collected no data on customers' total diets or total population impact.

Due to time and resource constraints, our data are unlikely to be representative. Furthermore, we did not reach data saturation in interviews, nor was our customer survey validated or piloted prior to use. Our findings may not be generalisable beyond the UK.

### Strengths and limitations in relation to other studies

There are a limited number of intervention studies targeting takeaways in England, and few have been evaluated.[23] Most interventions to date have been delivered by local authorities, limiting their geographical reach. Suppliers, such as HC, have a much greater geographical reach. While mandatory approaches to portion control may be more effective than voluntary schemes,[30 31] these may be harder to implement.[32 33] Our intervention is a rare voluntary, industry-led approach to portion control. Difficulties engaging independent takeaways in public health interventions have been previously described, where simply identifying a given takeaway owner can be challenging.[17] The 29% recruitment rate we achieved compares favourably with other interventions in the sector; in another, unpublished, local authority-led study, we achieved a 10% recruitment rate.[34] We received limited feedback from those who did not attend the engagement event, although lack of staff cover may be a problem.

### Study implications

Our findings suggest that, within the takeaway sector, it is feasible to develop a supplier-led intervention based on 'providing information' and 'enabling choice'[35] and that this is acceptable across stakeholders. We highlight the importance of product-specific packaging that constrains portion size, which can, in part, offset variability in servers' ability to deliver consistent portions. HC's smaller portion box packaging was designed and branded to deliver a smaller sized 'Lite-BITE' meal.[36] Sales of these boxes provide evidence of longer-term viability; in 2017, HC sold 552 300 units of the 'Lite-BITE' boxes to 253 unique accounts (D. McLean, personal communication, 2018). Takeaway owners/managers seem likely to be more receptive to messages about portion control from peers than external organisations, framed primarily in the context of the potential financial benefits. The individual responsible for implementing changes in a takeaway (usually the owner or manager) may require clear and practical instructions on how to make changes. Ideally, interventions should seek to engage with takeaway owners or those responsible for branding, and this person should communicate changes to serving staff, within the wider context of good 'customer service'.

### Unanswered questions and future research

HC is not planning further engagement events. An alternative platform to deliver the information and guidance in a collective format may be required to maximise the potential of smaller portion packaging (eg, seminars at trade events). While the promotion of smaller portion meals was broadly acceptable, over half of the customers surveyed, that had not previously purchased the smaller portion meals, were not interested in trying in the future. However, smaller portion meals were clearly attractive to others. Further work is required to assess whether and how customer choices can be further changed.[35] Qualitative interviews with customers could usefully inform this, inclusive of their views with regards to meal value-for-money. Future research could explore the impact of smaller portion meals in Fish & Chip shops on customers' diet and the wider public health implications, as well as the potential to promote smaller portion meals through trade organisations and their events. Owners and managers would additionally benefit from a clearer definition of what constitutes a smaller portion meal from a practical perspective. Defining and developing guidelines to support delivery would be of use.

While this intervention was feasible in a sample of Fish & Chip shops, further work should seek to identify other sectors of the takeaway and wider catering industry where such an approach could be applied.

**Author affiliations**
[1]Institute of Health and Society, Newcastle University, Newcastle upon Tyne, UK
[2]Human Nutrition Research Centre, Newcastle University, Newcastle upon Tyne, UK
[3]Fuse – UKCRC Centre for Translational Research in Public Health, Newcastle upon Tyne, UK
[4]Department of Sport and Exercise Sciences, Durham University, Durham City, UK

5Newcastle Business School, Northumbria University, Newcastle upon Tyne, UK
6Henry Colbeck Limited, Gateshead, UK
7MRC Epidemiology Unit, Centre for Diet and Activity Research (CEDAR), University of Cambridge, Cambridge, UK
8Department of Science, School of Science, Engineering and Design, Teesside University, Middlesbrough, UK

**Acknowledgements** The authors thank Duncan McLean (director) and Bill Colbeck (chairman) of Henry Colbeck Limited for their time and commitment of resources to delivering the intervention, as well as their patience, understanding and respect of the study's research process. They would also like thank Laura Cutler, Eimear Duffy, Joel Halligan, Eimear Mullen and Maisie Rowland from the Human Nutrition Research Centre, Newcastle University who provided invaluable help with data collection.

**Contributors** JA, VA-S, CDS, AAL, MWh and AJA: devised the concept for the Foodscape project. MWo: developed the intervention with support from LG, VAS and AJA. The evaluation study and associated methods were designed by LG, FHB, JA, VAS, LP, WW, CDS, MWh and AJA. Data collection was overseen by LG and completed by LG, MWo, FHB, NH and AJA. LG: led on data analysis and drafting the manuscript, supported by MWh and AA. All authors provided critical comments on drafts of the manuscript and read and approved the final version.

**Funding** This paper presents independent research funded by the National Institute for Health Research (NIHR) School for Public Health Research (SPHR). This research was part of the SPHR-funded project: Transforming the 'foodscape': development and feasibility testing of interventions to promote healthier takeaway, pub or restaurant food, with additional support from Durham and Newcastle Universities. SPHR is funded by the National Institute for Health Research (NIHR). At the time of funding, SPHR was a partnership between the Universities of Sheffield, Bristol, Cambridge, Exeter, University College London; The London School for Hygiene and Tropical Medicine; the LiLaC collaboration between the Universities of Liverpool and Lancaster; and Fuse, the Centre for Translational Research in Public Health, a collaboration between Newcastle, Durham, Northumbria, Sunderland and Teesside Universities. Authors LG, FHB, NH, VAS, LP, CDS, AAL and AJA are members of Fuse. Funding for Fuse comes from the British Heart Foundation, Cancer Research UK, Economic and Social Research Council, Medical Research Council and the National Institute for Health Research, under the auspices of the UK Clinical Research Collaboration, and is gratefully acknowledged. AA is funded by the NIHR as a NIHR Research Professor. JA and MWh are funded by the Centre for Diet and Activity Research (CEDAR), a UKCRC Public Health Research Centre of Excellence. Funding from the British Heart Foundation, Cancer Research UK, Economic and Social Research Council, Medical Research Council, the National Institute for Health Research and the Wellcome Trust, under the auspices of the UK Clinical Research Collaboration, is gratefully acknowledged.

**Disclaimer** The views expressed in this publication are those of the author(s) and not necessarily those of the NHS, NIHR or the Department of Health and Social Care. The SPHR was not involved in the design of the study, analysis, interpretation of data nor writing the manuscript.

**Competing interests** The research funding contributed to the cost of design and production of some of the intervention materials. HC also contributed to the costs of intervention development and funded delivery. HC developed and sell the 'Lite-BITE®' Fish & Chip meal boxes detailed in the study. Neither the project, nor individual researchers received financial contributions from HC for this study or any other work. MWh is funded by NIHR as Director of its Public Health Research Funding Programme. AJA is funded by NIHR as a Research Professor and the National Director of NIHR SPHR. CS and MWh are principal investigators in the NIHR SPHR. MWo led and delivered the intervention and provided the packaging materials; at the time of the research he was an undergraduate student at Newcastle Business School, Northumbria University on secondment to HC. After completion of the study (and his degree programme at Northumbria University), he became an employee of HC. HC an independent supplier to over 2500 Fish & Chip Shops in the North-East of England, Yorkshire, Cumbria and Scotland. HC is a private limited company founded in 1893 and based in Gateshead, North-East England. Company number 00822749 (www.colbeck.co.uk).

**Patient consent for publication** Not required.

**Provenance and peer review** Not commissioned; externally peer reviewed.

**Data sharing statement** Anonymised data relating to the covert observations, sales of Fish & Chip meals, customer survey and semi-structured interviews are available on reasonable request from the corresponding author (LG).

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
