## [Reviewer comments · BMJ Open]

ARTICLE DETAILS

TITLE (PROVISIONAL)	Feasibility of working with a wholesale supplier to co-design and test acceptability of an intervention to promote smaller portions: an uncontrolled before-and-after study in British Fish & Chip Shops
AUTHORS	Goffe, Louis; Hillier-Brown, Frances; Hildred, Natalie; Worsnop, Matthew; Adams, J; Araujo-Soares, Vera; Penn, Linda; Wrieden, Wendy; Summerbell, Carolyn; Lake, Amelia; White, Martin; Adamson, Ashley

VERSION 1 – REVIEW

REVIEWER	Reviewer name: Pierre Chandon Institution and Country: INSEAD, France Competing interests: None declared
REVIEW RETURNED	21-Jul-2018

GENERAL COMMENTS	Congratulations to the authors for developing this interesting and novel intervention. I totally agree that businesses and their suppliers can contribute to the fight against overeating, and your results show that it is the case. The paper repeatedly mentions that financial considerations were key in the owners' decision to accept to promote smaller portions. Yet, there is no information about the price, cost or margin of the regular and smaller portions. Similarly, the paper mentions collecting sales data, but they are not reported. I imagine that you are planning to keep them for a second paper. Personally, I would have preferred to have everything in one paper. Clearly, financial considerations are key to determine the "acceptability" of the intervention, which is your goal. Finally, I found the paper unnecessarily long and detailed. I think that you could cut it down easily from 40 to 25-30 pages or fewer without losing much information.
--

REVIEWER	Reviewer name: Eva Almiron-Roig Institution and Country: Centre for Nutrition Research, University of Navarra, Spain Competing interests: None declared
REVIEW RETURNED	17-Sep-2018

GENERAL COMMENTS	This manuscript describes the feasibility and initial results of a one-off, brief intervention to promote offerings of smaller portion sizes in Fish and Chips shops in the North-East of England (U.K.). While a few "downsizing" interventions have been described previously, these have tended to be in cafeteria or restaurant settings, and not so much in takeaway outlets. This is relevant as food sold in these locations is typically energy-dense, cheap and palatable, all of which encourages overconsumption.
---

Another important aspect of this research is that it involves all relevant stakeholders, such as wholesales suppliers, shop owners, managers and customers, while other research has tackled predominantly owners and managers or only customers. A logic model is presented which underpins the study design, as well as detailed documentation related to the development and implementation of the intervention. The exception to this is the customer survey and interview scripts (these two would have been helpful as part of the supplementary info).

While the sample size was small (12 shops), it looks acceptable for a feasibility study, especially considering the difficulties in engaging stakeholders in these type of studies. On the other hand I missed more detail on the reporting of the qualitative data, as well as the method for analysis. I am concerned that this part may have been slightly overlooked (please justify if otherwise). Equally, more clarity is needed in the quantitative results description.

I also missed more emphasis on a particular point that came out of this study, which is that, despite customers being aware of smaller portion size offerings, only 20% reported buying them, and only 46% reported intention to buy them in future. I believe this deserves attention as it may suggest that interventions that are merely informative are likely to have a small uptake by customers, although they may be attractive to owners. What should be done about this? (e.g. should PHE encourage posters with more impactful health warnings across health centres, community venues, etc. to link the portion size recommendations with actual consequences? Could the shop posters provide more contextual information on how the energy/fat provided compares with daily needs? Alternatively, could owners maintain benefits by adding vegetables, perhaps, to their recipes?).

Demonstrating to the owners how the current portion size fits in a regular plate is a good way to increase awareness and maybe messages should be more on that line. Many owners considered that they already offer small portions however given they could reduce further, my suspicion is that what they consider small portions are merely regular ones, and what they consider regular ones are actually oversized ones).

Once these limitations are addressed, I think this research contributes usefully to the literature by helping to identify and address problems in future interventions.

Specific comments:

ABSTRACT

L54 – data were (?)

L79 – study was UK based and targeting a typical UK type of food outlet, but probably adaptable to other countries

INTRODUCTION

This section is clear and to the point. I further recommend: Hollands et al. 20171, Steenhuis & Poelman 20172; Reinders et al. 20173, Hollands et al. 20184.

L88 – I would say that takeaways deliver excessive energy by means of large portion sizes (if that is sustained by ref. 1 in the manuscript).

L91-92 – Consider alternative/additional reference 5.

L98 – Estimated average daily energy requirement.

L109 – Fish & Chips Shops in the NE of England.

METHODS

L114-122 – I was surprised that HC accepted to do all the work and cover the costs at no direct/foreseeable immediate benefit.

Did they expect to gain good image related to social responsibility? (L370-372).

L129 – It seems that the use of the box is a key part of the intervention to restrict portion size. However later on (L265) it is stated that all but one shop already used a box as opposed to paper wrapping at baseline. Please comment on how this may have affected the outcome (in discussion if needed).

L135 – I missed info on the “Bio box” and “MK” options that appear in the pledge form.

L138 – From the text I understand that in addition to the HC staff, two shop owners also participated in the delivery of the intervention. Please give reasoning for this (later on you mention that receiving information from peers may be more effective than from others, was that the reason?).

L143 – What benefits in particular? Please give details.

L172 – There is an extra “.” after shops.

L182 – I would insert Figure 2 here. Otherwise the timings are not clear.

L195 – Please explain the purpose of completing and taking a picture of the pledge form.

L198 – It is not totally clear what the purpose of the covert observations was. Also, did owners know, and most importantly, consent, to these? Discuss any ethical implications.

L208 – It seems hard work for owners to have to keep sales records, except that they were already doing it? Please clarify.

L211 – Please provide the survey as supplementary information. Has this tool been validated, previously used, or piloted? Please provide details.

L218 – Was there any question about what customers thought of the price benefit, or their perceptions about shops offering smaller portions? Overall the survey seemed quite brief and more aspects could have been explored.

L220 – How many people took part in the interviews in total? (n=3?). Please provide script in the supplementary info.

L225 – Who conducted the interviews, were they properly trained?

L227 – Please expand on how the qualitative data analysis was done. For example, which steps were followed as part of the thematic framework analysis (these are well described in the literature, e.g.6), who coded the data, were they trained researchers? How were potential conflicts between coder’s perceptions or knowledge and code assignment dealt with? See appropriate literature, e.g.6–8. Please state if any software was employed.

For the quantitative data, please provide details of which descriptive statistics were used and the software.

RESULTS

TABLE 1 – Please provide full details in table legend. Currently it is non-informative. For example, are the numbers the sample sizes for each group? What is the Bio box offer? This is not explained in the text. Unclear what this has to do with portion size actually.

Line 256 – Only 2 shops clearly displayed availability of smaller portion meals at baseline, yet most owners perceived they already offered smaller portions. There seems to be a conflict between what health professionals interpret as a small portion size and what owners perceived it to be (as previously suggested9). I would also consider whether owner’s perceptions may be influenced by perceived longer-term financial benefits of offering larger portions.

TABLE 2 – Please label the data for regular vs. small portion meals in agreement with the nomenclature used in the text. As it stands it is not easy to interpret.

Consider presenting the data for weights for each type of meal in a bar chart.

L280 – Were these percentages significantly different?

L284 – I found this survey one of the most important components of the intervention, unfortunately the sample size is rather small. Also the data collected were rather restricted. Who were the 46% who said would be willing to try the smaller meals? (e.g. women, lean or overweight? SES level? This was not apparently collected yet it is relevant).

TABLE 3 – Some of the topics are unclear (e.g. for Portion Size, what was asked exactly?). Please include complete survey in supplementary info.

Overall the survey results seem to suggest that the poster intervention was not very effective on customer's choices (80% did not try smaller portions, yet >70% were aware of them). Although the paper focuses on the intervention implementation, I believe these results merit more attention to understand the real impact of the intervention.

L293 and further – The reporting of the qualitative data is a bit poor in my opinion. There are very few quotes, without identification (e.g. male, 56 y old) and some of the themes seem very general (e.g. "posters and shop setting", "customer feedback"). I would suggest checking the coding matrix with a (additional) qualitative data expert in case it can be improved.

L330 – It is interesting that a group of builders (presumably men?) were interested in the small portion offers. I think this deserves expanding given this group may be more frequent consumers and less research has been done on men. This should be related to the respondent's demographic data (e.g. BMI, age).

L333 – Unclear (as only limited information is provided on how the recording was done).

L360 – responses were positive.

DISCUSSION

The discussion is well structured and easy to read. It misses/lacks emphasis on some key points though.

For the statement of main findings I would consider mentioning, in addition to what is said, that although the intervention did work amongst the food shops and appeared to be acceptable to customers, it was not impactful, from the reported low uptake (only 20% bought small meals despite high awareness; about half still would not purchase small portion sizes in future). Clearly, other/improved strategies need to be considered for more impact. It is also important to note that while the wholesaler reported big sales of the small portion size boxes, they did not plan to engage further, confirming a lack of sustainability for the intervention in its current format.

The 46 customers surveyed probably were not representative and the survey did not explore in sufficient detail reasons for the low uptake. That only 20% had purchased smaller meals suggests an absence of perceived need or perceived benefit for these meals. What are the reasons behind this? (taste, price, etc.?).

Did the intervention represent a financial challenge to shop owners? To what extent could this have affected how much they decided to engage?

Why was the pledge unsuccessful? Has it worked in other contexts?

L407 - 408 - Noting the IMD decile row in Table 1, there were only 3 shops with lower deciles. Probably a more balanced distribution of shop locations would have helped?

L412 – What other marketing materials?

	L423 – Ref 29. For the UK Responsibility Deal this paper may be (more) useful¹⁰. L426 – Ref 30 is not correctly cited. If not published this should go in the text as “unpublished data”. Consider giving the actual recruitment rate in this other source. L435 – I believe personal communications need to have a name and a date? (please check with journal guidelines). AUTHOR CONTRIBUTIONS L517 - Please specify who carried out the quantitative and/or qualitative data analysis. Ideally these should not be carried out by members from or persons with an interest in, the commercial partner (including on secondment). References  1. Hollands GJ, Bignardi G, Johnston M, et al. The TIPPME intervention typology for changing environments to change behaviour. Nat Hum Behav. 2017;1(8):0140. 2. Steenhuis I, Poelman M. Portion Size: Latest Developments and Interventions. Curr Obes Rep. 2017;6(1):10-17. 3. Reinders MJ, Huitink M, Dijkstra SC, Maaskant AJ, Heijnen J. Menu-engineering in restaurants - adapting portion sizes on plates to enhance vegetable consumption: A real-life experiment. Int J Behav Nutr Phys Act. 2017;14(1):41. 4. Hollands GJ, Cartwright E, Pilling M, et al. Impact of reducing portion sizes in worksite cafeterias: A stepped wedge randomised controlled pilot trial. Int J Behav Nutr Phys Act. 2018;15(1):78. 5. Burgoine T, Forouhi NG, Griffin SJ, Wareham NJ, Monsivais P. Associations between exposure to takeaway food outlets, takeaway food consumption, and body weight in Cambridgeshire, UK: population based, cross sectional study. BMJ. 2014;348:g1464. 6. Braun V, Clarke V. Using thematic analysis in psychology. Qual Res Psychol. 2006;3:77-101. 7. Hsieh H-F, Shannon SE. Three Approaches to Qualitative Content Analysis. Qual Health Res. 2005;15(9):1277-1288. 8. Draper A, Swift JA. Qualitative research in nutrition and dietetics: data collection issues. J Hum Nutr Diet. 2011;24(1):3-12. 9. Lewis HB, Forwood SE, Ahern AL, et al. Personal and social norms for food portion sizes in lean and obese adults. Int J Obes. 2015;39(8):1319-1324. 10. Knai C, Petticrew M, Durand MA, et al. Has a public–private partnership resulted in action on healthier diets in England? An analysis of the Public Health Responsibility Deal food pledges. Food Policy. 2015;54:1-10.
--	--

REVIEWER	Reviewer name: Dr Rajshri Roy Institution and Country: University of Auckland Competing interests: None Declared
REVIEW RETURNED	19-Sep-2018

GENERAL COMMENTS	Despite the fact that a broad range of interventions aimed at portion size has been suggested in the literature, very few intervention studies aimed at portion size have been conducted thus far, therefore, this research assessing the feasibility of portion size intervention at fish and chips outlets in Northern England is warranted. Upon reading, I believe the manuscript will improve from the following revisions:
--

	1. It would be valuable to provide the readers with detailed information about the current fish and chips market by researching trends and analyzing the market and using that in the introduction and to support the discussion of the results. 2. A table could be included summarising all the measures used and time of measurement. This would make the primary measures of acceptability and feasibility of this portion size intervention procedures more clear. 3. The manuscript can be organised better to focus on the process of the intervention and how it can be adapted by others and in other settings. The procedures and measures in the methods section should focus more on outcomes of the study. 4. In the methods section, there could be more emphasis on explaining the resources used & management of the study as this is a feasibility study that manages multiple stakeholders, it would be good to get that details in the methods if other researchers want to replicate the study in other settings. 5. In the results section, there is segregated results sections about surveys, food sales and interviews. There should be a detailed section on intervention acceptability. The authors talk about the intervention being acceptable but there is no proper discussion about the acceptability of the intervention. This could be improved by doing the following: - Evaluation of participant responses - it would be very valuable if some cross-tabulations could be performed perhaps using chi-square to see if any significant differences were found in responses by age, gender, or ethnicity etc.- Chi-square or other statistical tests could be used to analyze the food sales data. Were there any significant differences observed in sales between sites before the intervention? or between pre intervention and intervention periods at the sites? It's been a while since the study was conducted. Is the portion size intervention still in place, if so, can we determine whether sales changed over time at the intervention sites? 6. The authors explained the limitations of their study in detail, however, for a feasibility study it would be worthwhile to know more about what the limitations of the actual intervention were and how future studies can improve the intervention. What should we do in the future? What did you learn from the feasibility of the intervention research that you would do differently for the larger study? What do you recommend future researchers to focus on when it comes to conducting such portion size interventions? 7. The researchers should identify strategies in the manuscript that needs to be addressed, note the challenges they faced and/or revise components of the intervention prior to designing a pilot study to more formally evaluate intervention outcomes. 8. The manuscript should be revised and focus on the process of developing and implementing this intervention in detail and result in preliminary examination of participant responses to the intervention.
--	--

REVIEWER	Reviewer name: Isabelle Szmigin Institution and Country: University of Birmingham, UK Competing interests: None
REVIEW RETURNED	26-Sep-2018

GENERAL COMMENTS	I have assessed this paper on the basis of its objectives, i.e. to be an intervention. I have a few points to make and then some additional elements which I think could improve the impact of the paper. The paper is well written and presented. It is a shame there are so few consumer surveys and that they are very brief (although this is understandable). It might be worth considering in the future some qualitative work with consumers about why they buy fish and chips etc as discussed further below. Ethics. In the paper the authors are clear that the observation was covert but they refer to it only as observation in the ethical part; should they not have told shop owners that covert observation would take place? If they did they that should be in the paper. Early on in the paper the authors refer to value for money being an issue in buying larger portion sizes. However, this is not addressed in the survey and most say their reasons for buying were taste and convenience, only 3 mentioned price. Perhaps value for money should have been in the question but in terms of further research exploring the why of consumers is important and this could be included in the conclusions. I think more might be said about the problems of doing this kind of intervention and perhaps a bit more on why shop owners did not engage with the intervention. Finally it sounds like there are some practical issues around smaller sizes - having the right packaging but also it sounds like the size of the fish portions is also an issue - it might be worth suggesting further analysis of this as well as the value for money motivation.
---

VERSION 1 – AUTHOR RESPONSE

Reviewer: 1

Reviewer Name: Pierre Chandon

Institution and Country: INSEAD, France

Please state any competing interests or state 'None declared': None declared

Congratulations to the authors for developing this interesting and novel intervention. I totally agree that businesses and their suppliers can contribute to the fight against overeating, and your results show that it is the case.

The paper repeatedly mentions that financial considerations were key in the owners' decision to accept to promote smaller portions. Yet, there is no information about the price, cost or margin of the regular and smaller portions.

Response: We have added details regarding the cost of both the regular and smaller portion meals. We do not have figures relating margin. Please see Table 2, page 17, line 295.

Similarly, the paper mentions collecting sales data, but they are not reported. I imagine that you are planning to keep them for a second paper. Personally, I would have preferred to have everything in one paper.

Due to the inconsistencies with regards to the number of days each shop reported sales of both regular and small portion meals, we decided that reporting the proportion of smaller portions sold across all shops was the most suitable metric for this feasibility study. Please see our clarification in the subsection 'Sales of Fish & Chip meals' in results, page 17, line 300.

Clearly, financial considerations are key to determine the "acceptability" of the intervention, which is your goal.

There were no direct financial costs directly from implementing the intervention and we have clarified this. Please see page 8, line 136.

Finally, I found the paper unnecessarily long and detailed. I think that you could cut it down easily from 40 to 25-30 pages or fewer without losing much information.

Response: We have reviewed and identified text throughout the manuscript that we could suitably trim.

Reviewer: 2

Reviewer Name: Eva Almiron-Roig

Institution and Country: Centre for Nutrition Research, University of Navarra, Spain

Reviewer: Eva Almiron-Roig, University of Navarra, Spain

Review date: 17th September 2018

General comments:

This manuscript describes the feasibility and initial results of a one-off, brief intervention to promote offerings of smaller portion sizes in Fish and Chips shops in the North-East of England (U.K.). While a few "downsizing" interventions have been described previously, these have tended to be in cafeteria or restaurant settings, and not so much in takeaway outlets. This is relevant as food sold in these locations is typically energy-dense, cheap and palatable, all of which encourages overconsumption. Another important aspect of this research is that it involves all relevant stakeholders, such as wholesales suppliers, shop owners, managers and customers, while other research has tackled predominantly owners and managers or only customers. A logic model is presented which underpins the study design, as well as detailed documentation related to the development and implementation of the intervention. The exception to this is the customer survey and interview scripts (these two would have been helpful as part of the supplementary info).

While the sample size was small (12 shops), it looks acceptable for a feasibility study, especially considering the difficulties in engaging stakeholders in these type of studies. On the other hand I missed more detail on the reporting of the qualitative data, as well as the method for analysis. I am concerned that this part may have been slightly overlooked (please justify if otherwise). Equally, more clarity is needed in the quantitative results description.

I also missed more emphasis on a particular point that came out of this study, which is that, despite customers being aware of smaller portion size offerings, only 20% reported buying them, and only 46% reported intention to buy them in future. I believe this deserves attention as it may suggest that interventions that are merely informative are likely to have a small uptake by customers, although they may be attractive to owners. What should be done about this? (e.g. should PHE encourage posters with more impactful health warnings across health centres, community venues, etc.

to link the portion size recommendations with actual consequences? Could the shop posters provide more contextual information on how the energy/fat provided compares with daily needs? Alternatively, could owners maintain benefits by adding vegetables, perhaps, to their recipes?).

Demonstrating to the owners how the current portion size fits in a regular plate is a good way to increase awareness and maybe messages should be more on that line. Many owners considered that they already offer small portions however given they could reduce further, my suspicion is that what they consider small portions are merely regular ones, and what they consider regular ones are actually oversized ones).

Once these limitations are addressed, I think this research contributes usefully to the literature by helping to identify and address problems in future interventions.

Thank you for your detailed and helpful comments. Please see below for our responses and revisions to our manuscript.

Specific comments:

ABSTRACT

L54 – data were (?)

We have corrected as suggested. Please see page 4, line 55.

L79 – study was UK based and targeting a typical UK type of food outlet, but probably adaptable to other countries

We have stated that our findings may not be generalizable beyond the UK. Please see page 24, line 450.

INTRODUCTION

This section is clear and to the point. I further recommend: Hollands et al. 20171, Steenhuis & Poelman 20172; Reinders et al. 20173, Hollands et al. 20184.

Thank you for these additional references. We have added Steenhuis & Poelman (2017) and Hollands et al. (2018) to our manuscript. Please see page 6, line 101.

L88 – I would say that takeaways deliver excessive energy by means of large portion sizes (if that is sustained by ref. 1 in the manuscript).

Thank you for refining this point. Yes, it is sustained by reference 1. We have revised accordingly. Please see page 6, line 89.

L91-92 – Consider alternative/additional reference5.

We are aware of this reference. However, Burgoine et al.'s study focus is on spatial exposure, while related, the existing references are more suited to the subject of this sentence.

L98 – Estimated average daily energy requirement.

We have revised as suggested. Please see page 6, line 100.

L109 – Fish & Chips Shops in the NE of England.

We have revised as suggested. Please see page 7, line 112.

METHODS

L114-122 – I was surprised that HC accepted to do all the work and cover the costs at no direct/foreseeable immediate benefit. Did they expect to gain good image related to social responsibility? (L370-372).

We explored this in the interviews with the intervention deliverers, as reported in subsection 'Intervention deliverers' motivations regarding intervention delivery'. We found that the primary motivation was to drive innovation and support Fish & Chip Shops to remain competitive in a diverse takeaway marketplace. Social responsibility was not mentioned as a motivation.

L129 – It seems that the use of the box is a key part of the intervention to restrict portion size. However later on (L265) it is stated that all but one shop already used a box as opposed to paper wrapping at baseline. Please comment on how this may have affected the outcome (in discussion if needed).

Thank you for providing the opportunity to clarify this point. We have added further detail to our discussion. Please see page 23, line 424.

L135 – I missed info on the "Bio box" and "MK" options that appear in the pledge form.

We have added a packaging key to the supplementary file. Please see page 2 of supplementary file B (goal-setting form).

L138 – From the text I understand that in addition to the HC staff, two shop owners also participated in the delivery of the intervention. Please give reasoning for this (later on you mention that receiving information from peers may be more effective than from others, was that the reason?).

We have added detail of the description of the intervention's co-design. Please see page 8, line 138.

L143 – What benefits in particular? Please give details.

We have added examples. Please see page 8, line 152.

L172 – There is an extra "." after shops.

It is grammatically correct. The colon is there to signify the start of a conditional list.

L182 – I would insert Figure 2 here. Otherwise the timings are not clear.

We respectfully disagree. Figure 2 contains results and is appropriately placed within the manuscript.

L195 – Please explain the purpose of completing and taking a picture of the pledge form.

This was for data collection purposes, to record pledged changes. We have added this detail. Please see page 11, line 207.

L198 – It is not totally clear what the purpose of the covert observations was. Also, did owners know, and most importantly, consent, to these? Discuss any ethical implications.

The purpose was to provide objective measures of change. Please see page 11, line 210. These observations were made in a public place with no recording of individuals nor their behaviour. Additionally, we have not reported information that would enable individual shop identification. As such, no ethical considerations were raised.

L208 – It seems hard work for owners to have to keep sales records, except that they were already doing it? Please clarify.

We have clarified that data was not sufficient for detailed analysis. Please see page 17, line 300.

L211 – Please provide the survey as supplementary information. Has this tool been validated, previously used, or piloted? Please provide details.

We have added this tool as a supplementary document. It had not been validated or previously piloted, we have detailed this as a limitation. Please see page 24, line 449.

L218 – Was there any question about what customers thought of the price benefit, or their perceptions about shops offering smaller portions? Overall the survey seemed quite brief and more aspects could have been explored.

We did not ask customers their thoughts on the price benefits, or their more detailed perceptions regarding smaller portions. The survey was confined to the brief time period between a customer ordering their food and when it was served by shop staff. We have both acknowledged this as a limitation (please see page 24, line 441) and as an area for future work (please see page 26, line 488).

L220 – How many people took part in the interviews in total? (n=3?). Please provide script in the supplementary info.

We have added the interview topic guides as supplementary material. The number of people that took part in the interviews is appropriately included in the results section.

L225 – Who conducted the interviews, were they properly trained?

We have added author information to the methods. Both LG and FHB are suitably trained. Please see page 12, lines 236 & 239.

L227 – Please expand on how the qualitative data analysis was done. For example, which steps were followed as part of the thematic framework analysis (these are well described in the literature, e.g.6), who coded the data, were they trained researchers? How were potential conflicts between coder's perceptions or knowledge and code assignment dealt with? See appropriate literature, e.g.6–8. Please state if any software was employed.

We have included further detail. This work was led by LG, who is trained and previously led and published qualitative research (see below). Please see the subsection 'data analysis', page 12, line 241. Despite this, due to the limited sample size, data saturation was not met and we have acknowledged this in our limitations. Please see page 24, line 448.

Goffe et al. 2018. The challenges of interventions to promote healthier food in independent takeaways in England: qualitative study of intervention deliverers' views. BMC Public Health 18:184.

For the quantitative data, please provide details of which descriptive statistics were used and the software.

We have modified as suggested and noted that all quantitative work was carried out in R. Please see subsection 'data analysis', page 12, line 242.

RESULTS

TABLE 1 – Please provide full details in table legend. Currently it is non-informative. For example, are the numbers the sample sizes for each group? What is the Bio box offer? This is not explained in the text. Unclear what this has to do with portion size actually.

We have modified Table 1 to clear any confusion. Bio boxes are describe in a key at the bottom of the table. Please see Table 1, page 14, line 272.

Line 256 – Only 2 shops clearly displayed availability of smaller portion meals at baseline, yet most owners perceived they already offered smaller portions. There seems to be a conflict between what health professionals interpret as a small portion size and what owners perceived it to be (as previously suggested⁹). I would also consider whether owner’s perceptions may be influenced by perceived longer-term financial benefits of offering larger portions.

Respectfully, this is not an issue of individual perception as investigated by Lewis et al (2015). The purpose of the covert observations was to collect objective measures. Please see our clarification on page 11, line 210. Self-reporting of smaller portion meals by either the owner or manager is not confirmation of the clear sign-posting of smaller portion meals. Just because they told us that they offer smaller portion meals that does not mean that such meals are obvious to customers when they enter the shop and view menu information. As detailed in our study’s title and objective, it was about the overt promotion of smaller portions and not simply availability.

TABLE 2 – Please label the data for regular vs. small portion meals in agreement with the nomenclature used in the text. As it stands it is not easy to interpret.

We have updated as suggested. Please see Table 2, page 17, line 295.

Consider presenting the data for weights for each type of meal in a bar chart.

We considered alternative ways to display the data. However, as it was a feasibility study and the data is under-powered, we felt to present the data in graphical form could be misleading.

L280 – Were these percentages significantly different?

It was not appropriate to apply statistical tests, as this was not an effectiveness trial. We have added further detail in our methods that we did not make any statistical inferences with respect to the data we collected. Please see page 12, line 243.

L284 – I found this survey one of the most important components of the intervention, unfortunately the sample size is rather small. Also the data collected were rather restricted. Who were the 46% who said would be willing to try the smaller meals? (e.g. women, lean or overweight? SES level? This was not apparently collected yet it is relevant).

Please see previous comments. Additionally, please see detail about when customers were surveyed. It was not feasible to collect data relating to their weight or socio-economic status during their wait for their food.

TABLE 3 – Some of the topics are unclear (e.g. for Portion Size, what was asked exactly?). Please include complete survey in supplementary info.

We have included the survey questions as supplementary material.

Overall the survey results seem to suggest that the poster intervention was not very effective on customer’s choices (80% did not try smaller portions, yet >70% were aware of them). Although the paper focuses on the intervention implementation, I believe these results merit more attention to understand the real impact of the intervention.

Thank for raising this important point. We have reflected on this point in suggestions for future research. Please see page 26, line 485.

L293 and further – The reporting of the qualitative data is a bit poor in my opinion. There are very few quotes, without identification (e.g. male, 56 y old) and some of the themes seem very general r (e.g. “posters and shop setting”, “customer feedback”). I would suggest checking the coding matrix with a (additional) qualitative data expert in case it can be improved.

We have added further detail in the methods. Respectfully, manuscripts submitted to this journal have a restricted word limit. This is a mixed methods study and we had to select the data that was most pertinent to the study's objectives. We have linked the quotes from the owners and manager to their respective business (ID01-ID12). This was not possible with those that delivered the intervention as this would compromise anonymity. Please see results, subsection ‘interviews with owners/managers’, page 19, line 315.

L330 – It is interesting that a group of builders (presumably men?) were interested in the small portion offers. I think this deserves expanding given this group may be more frequent consumers and less research has been done on men. This should be related to the respondent's demographic data (e.g. BMI, age).

Our previously published work on a UK population-representative cohort did not find a gender difference with regards to frequency of consumption of takeaway meals at home (Adams et al. 2015). While not appropriate to comment on findings within the results, we have drawn upon this example in the discussion. Please see page 26, line 487. While this cited example may indicate the wide range of people potentially interested in the smaller portion meals, we must be balanced in not concluding too much with regards to specific demographic groups.

L333 – Unclear (as only limited information is provided on how the recording was done).

Please see ‘sales of Fish & Chip meals’ in the results section (page 17, line 297). Here we detailed how they return the sales data to the research team. In ‘the ease of recording sales data’ is how they completed this task (page 20, line 356).

L360 – responses were positive.

Thank you for identifying this grammatical mistake. We have corrected. Please see page 21, line 384.

DISCUSSION

The discussion is well structured and easy to read. It misses/lacks emphasis on some key points though.

For the statement of main findings I would consider mentioning, in addition to what is said, that although the intervention did work amongst the food shops and appeared to be acceptable to customers, it was not impactful, from the reported low uptake (only 20% bought small meals despite high awareness; about half still would not purchase small portion sizes in future). Clearly, other/improved strategies need to be considered for more impact.

This was not a study to test effectiveness. It was to test and assess acceptability and feasibility, inclusive of data collection methods. We have modified our aim to explicitly state this, please see page 7, line 113. Therefore, it is not appropriate to state either way its impact. As previously suggested we have added addition suggestions related to this in the section ‘unanswered questions and future research’, please see page 26, line 481.

It is also important to note that while the wholesaler reported big sales of the small portion size boxes, they did not plan to engage further, confirming a lack of sustainability for the intervention in its current format.

This is correct and refers specifically to the method of engagement with regards to the 'engagement event'. However, the wider issue of promotion of smaller portion meals was viewed by the wholesale supplier as sustainable. This point is addressed in the subsection 'strengths and limitations of the study' in our discussion, please see page 24, line 429.

The 46 customers surveyed probably were not representative and the survey did not explore in sufficient detail reasons for the low uptake. That only 20% had purchased smaller meals suggests an absence of perceived need or perceived benefit for these meals. What are the reasons behind this? (taste, price, etc.?).

We have noted that sample was not representative in our discussion, please see page 24, line 448. The customer survey was useful in providing this initial insight, but we do not know why certain customers (and potentially different demographic groups) did not choose the smaller portion meals, nor was this explored. Thank you for highlighting this important issue and we have suggested that further work could be done to explore the reasons behind this. Please see page 26, line 487.

Did the intervention represent a financial challenge to shop owners? To what extent could this have affected how much they decided to engage?

There were no direct financial implications to Fish & Chip Shop participation in the study. As stated in the results section 'relationship between owner/manager and Henry Colbeck Limited' that while they appreciated the incentives on offer, they were not deemed necessary, please see page 19, line 319. The closest detailed financial issue was the one owner who detailed poor customer service being behind the reason for not offering the smaller portion meals in the evening, where the shop's staff could not be relied upon to deliver a smaller portion meal consistently, please see page 20, line 347.

Why was the pledge unsuccessful? Has it worked in other contexts?

Goal setting and action planning is an established technique in behavioural science. We are not aware of it being applied in other intervention within this same setting outside of our allied study, which is as yet unpublished. While the participants in our study stated they did not find the public pledge useful, as we were not evaluating impact we cannot state that they were unsuccessful.

L407 - 408 - Noting the IMD decile row in Table 1, there were only 3 shops with lower deciles. Probably a more balanced distribution of shop locations would have helped?

We would contend for a feasibility study, we obtain a good spread of Fish & Chip Shops across the deciles. As pointed out we had three from the three lowest deciles, and respectively four from the three highest deciles. There was also an exact 50/50 split between the top and bottom five deciles.

L412 – What other marketing materials?

Thank you, we have referred back to those identified by the owners/managers in our qualitative results and provided an example. Please see page 24, line 441.

L423 – Ref 29. For the UK Responsibility Deal this paper may be (more) useful¹⁰.

Thank you for the reference and we have added to our manuscript. Please see page 25, line 457.

L426 – Ref 30 is not correctly cited. If not published this should go in the text as "unpublished data". Consider giving the actual recruitment rate in this other source.

Thank you for highlighting, we have revised accordingly. Please see page 25, line 461.

L435 – I believe personal communications need to have a name and a date? (please check with journal guidelines).

Thank you for highlighting, we have revised accordingly. Please see page 25, line 472.

References

1. Hollands GJ, Bignardi G, Johnston M, et al. The TIPPME intervention typology for changing environments to change behaviour. *Nat Hum Behav.* 2017;1(8):0140.
2. Steenhuis I, Poelman M. Portion Size: Latest Developments and Interventions. *Curr Obes Rep.* 2017;6(1):10-17.
3. Reinders MJ, Huitink M, Dijkstra SC, Maaskant AJ, Heijnen J. Menu-engineering in restaurants - adapting portion sizes on plates to enhance vegetable consumption: A real-life experiment. *Int J Behav Nutr Phys Act.* 2017;14(1):41.
4. Hollands GJ, Cartwright E, Pilling M, et al. Impact of reducing portion sizes in worksite cafeterias: A stepped wedge randomised controlled pilot trial. *Int J Behav Nutr Phys Act.* 2018;15(1):78.
5. Burgoine T, Forouhi NG, Griffin SJ, Wareham NJ, Monsivais P. Associations between exposure to takeaway food outlets, takeaway food consumption, and body weight in Cambridgeshire, UK: population based, cross sectional study. *BMJ.* 2014;348:g1464.
6. Braun V, Clarke V. Using thematic analysis in psychology. *Qual Res Psychol.* 2006;3:77-101.
7. Hsieh H-F, Shannon SE. Three Approaches to Qualitative Content Analysis. *Qual Health Res.* 2005;15(9):1277-1288.
8. Draper A, Swift JA. Qualitative research in nutrition and dietetics: data collection issues. *J Hum Nutr Diet.* 2011;24(1):3-12.
9. Lewis HB, Forwood SE, Ahern AL, et al. Personal and social norms for food portion sizes in lean and obese adults. *Int J Obes.* 2015;39(8):1319-1324.
10. Knai C, Petticrew M, Durand MA, et al. Has a public-private partnership resulted in action on healthier diets in England? An analysis of the Public Health Responsibility Deal food pledges. *Food Policy.* 2015;54:1-10.

AUTHOR CONTRIBUTIONS

L517 - Please specify who carried out the quantitative and/or qualitative data analysis. Ideally these should not be carried out by members from or persons with an interest in, the commercial partner (including on secondment).

This information is contained in the 'author contributions' section. We have also added the relevant authors' initials to the subsection 'data analysis' in the methods, please see page 12, line 241. Author MWO was not involved in either analysis or interpretation of the data.

Reviewer: 3

Reviewer Name: Dr Rajshri Roy

Institution and Country: University of Auckland

Despite the fact that a broad range of interventions aimed at portion size has been suggested in the literature, very few intervention studies aimed at portion size have been conducted thus far, therefore, this research assessing the feasibility of portion size intervention at fish and chips outlets in Northern England is warranted. Upon reading, I believe the manuscript will improve from the following revisions:

1. It would be valuable to provide the readers with detailed information about the current fish and chips market by researching trends and analyzing the market and using that in the introduction and to support the discussion of the results.

An estimation of the total number of Fish & Chip Shops is in the introduction. Due to the word limit in the manuscript it was not possible expand further.

2. A table could be included summarising all the measures used and time of measurement. This would make the primary measures of acceptability and feasibility of this portion size intervention procedures more clear.

Please see Figure 2. This figure shows the study's process and data collected throughout.

3. The manuscript can be organised better to focus on the process of the intervention and how it can be adapted by others and in other settings.

This is the stated purpose of a template for intervention description and replication (TIDieR) checklist. As stated we completed a fully detailed TIDieR checklist which is included as Supplementary File C.

4. The procedures and measures in the methods section should focus more on outcomes of the study.

Respectfully, this is a feasibility and acceptability study. Their focus is to understand these two concepts and their suitability for an expanded trial. We have clarified that it was not an outcome evaluation trial in our study's aim. Please see page 7, line 113.

5. In the methods section, there could be more emphasis on explaining the resources used & management of the study as this is a feasibility study that manages multiple stakeholders, it would be good to get that details in the methods if other researchers want to replicate the study in other settings.

Again, please refer to Supplementary File C, the TIDieR checklist. This contains the requisite information: materials, procedures and who provided.

6. In the results section, there is segregated results sections about surveys, food sales and interviews. There should be a detailed section on intervention acceptability. The authors talk about the intervention being acceptable but there is no proper discussion about the acceptability of the intervention. This could be improved by doing the following:

a. Evaluation of participant responses - it would be very valuable if some cross-tabulations could be performed perhaps using chi-square to see if any significant differences were found in responses by age, gender, or ethnicity etc.

b. Chi-square or other statistical tests could be used to analyze the food sales data. Were there any significant differences observed in sales between sites before the intervention? Or between pre intervention and intervention periods at the sites? It's been a while since the study was conducted. Is the portion size intervention still in place, if so, can we determine whether sales changed over time at the intervention sites?

Test for statistical significance are not appropriate for this study. Please see the clarification in our study's aim (page 7, line 113). We did not design this data with statistical significance testing in mind, nor do we claim or infer that our findings are significant, as we were not evaluating impact. The purpose of the survey was to understand if this data collection method and questions were suitable for an expanded trial and to better understand what issues we would need to further explore. We have expanded our section 'unanswered questions and future research' based on the survey. Please see

page 26, line 485. Furthermore, there is qualitative information on acceptability emerging from the semi-structured interviews.

7. The authors explained the limitations of their study in detail, however, for a feasibility study it would be worthwhile to know more about what the limitations of the actual intervention were and how future studies can improve the intervention.

As stated this was not an evaluation of the effectiveness of the intervention nor the process. Please see our objectives in the abstract and the clarification in our study's aim in the introduction. Despite this, we do also comment on specific issues learnt through this study that could potentially improve the intervention. Please see the section 'study implication' where we detail the importance of product specific packaging.

8. What should we do in the future?

We have expanded our existing section in the discussion 'unanswered questions and future research', please see page 26, line 481.

9. What did you learn from the feasibility of the intervention research that you would do differently for the larger study?

Please see the section 'study implications' in the discussion. Please see page 25, line 465.

10. What do you recommend future researchers to focus on when it comes to conducting such portion size interventions?

This is a challenging question, which we do not think we can adequately answer, nor is appropriate, from our small-scale feasibility study. However, we have detailed the implications (page 25, line 465) within our specific setting, which may be useful to those working on interventions in related out-of-home food settings.

11. The researchers should identify strategies in the manuscript that needs to be addressed, note the challenges they faced and/or revise components of the intervention prior to designing a pilot study to more formally evaluate intervention outcomes.

This reflective information is contained within our structured discussion (Docherty & Smith 1999), most notably in the subsections 'study implications' and 'unanswered questions and future research'. We have added additional detail with regards to further work required to defining and the practical implementation of smaller portion meals.

12. The manuscript should be revised and focus on the process of developing and implementing this intervention in detail and result in preliminary examination of participant responses to the intervention.

Respectfully, as now explicitly stated in our introduction, this was not a process or outcome evaluation study (page 7, line 113). While this would be of value, and we have certainly commented on such issues in our discussion, our primary focus was the feasibility of working with a wholesale supplier and the acceptability of the intervention to shop owners/managers and their customers.

Reviewer: 4

Reviewer Name: Isabelle Szmigin

Institution and Country: University of Birmingham, UK

Comments: I have assessed this paper on the basis of its objectives, i.e. to be an intervention. I have a few points to make and then some additional elements which I think could improve the impact of the paper.

The paper is well written and presented. It is a shame there are so few consumer surveys and that they are very brief (although this is understandable). It might be worth considering in the future some qualitative work with consumers about why they buy fish and chips etc as discussed further below.

Response: Thank you for this suggestion, we have added it to the section 'unanswered questions and future research'. Please see page 26, line 488.

Comments: Ethics. In the paper the authors are clear that the observation was covert but they refer to it only as observation in the ethical part; should they not have told shop owners that covert observation would take place? If they did they that should be in the paper.

Response: We did not inform takeaway owners that we were carrying out covert, 'secret shopper' style, observational data collection. This is detailed in the ethics statement section which follows the discussion. It was decided that we did not require consent from owners/managers as we were not carrying out data collection beyond that of typical customer behaviour, nor would we be reported any information that would enable the identification of a specific Fish & Chip Shop.

Early on in the paper the authors refer to value for money being an issue in buying larger portion sizes. However, this is not addressed in the survey and most say their reasons for buying were taste and convenience, only 3 mentioned price. Perhaps value for money should have been in the question but in terms of further research exploring the why of consumers is important and this could be included in the conclusions.

This is a good point. While our customer survey did potentially identify customers' reasons for selecting the specific Fish & Chip Shop, it did not explore further reasons behind issues relating to value of the meal purchased. While resource limitation meant that we were unable to deliver detailed qualitative interviewing during this study. We have added this as one of our study's limitations. Please see page 24, line 441. It is also good suggestion for future work. As such we have updated the appropriate section. Please see page 26, line 487.

I think more might be said about the problems of doing this kind of intervention and perhaps a bit more on why shop owners did not engage with the intervention.

We have cited our previous qualitative study that explored the challenges of interventions in this setting with intervention deliverers and added further detail. Please see page 25, line 459. We have also expanded on recruitment and engagement as a limitation of our study. Please see page 25, line 462.

Comments: Finally it sounds like there are some practical issues around smaller sizes - having the right packaging but also it sounds like the size of the fish portions is also an issue - it might be worth suggesting further analysis of this as well as the value for money motivation.

Response: We have added suggestions for further qualitative work with customers. Please see page 26, page 487. Also we have add suggestions for further work with regards to the practicalities of delivering smaller portion meals. Please see page 26, line 492.

VERSION 2 – REVIEW

REVIEWER	Reviewer name: Pierre Chandon Institution and Country: INSEAD, France Competing interests: None declared
REVIEW RETURNED	11-Dec-2018

GENERAL COMMENTS	Congratulations to the authors for a well-done revision. You have answered all my questions. I hope that you will pursue this line of research and test the outcome of this intervention, which we now know to be feasible.
---

REVIEWER	Reviewer name: Eva Almiron-Roig Institution and Country: University of Navarra Competing interests: None declared
REVIEW RETURNED	19-Dec-2018

GENERAL COMMENTS	Thank you for your thorough revision and detailed response to my comments. I feel your updated version covers my points well. I would only suggest to add in the Limitations section of the Discussion a very short justification (as explained in your letter of response) to why the customer data collection could not be more extensive. E.g. in Line 441 you could say something on these lines (please modify as suitable): "Due to the timing of the survey administration, the customer survey had to be brief and it was not practical to conduct in-depth interviews with customers....". I think this way the reader will understand better the choice of data collection method. In my opinion all other comments are appropriately responded to/addressed.
--

REVIEWER	Reviewer name: Dr Rajshri Roy Institution and Country: The University of Auckland, New Zealand Competing interests: None
REVIEW RETURNED	29-Nov-2018

GENERAL COMMENTS	The authors have responded well to the critiques. They have addressed my comments and those of the other reviewers. The authors have incorporated some comments from the first round of review.
---

REVIEWER	Reviewer name: Isabelle Szmigin Institution and Country: University of Birmingham, UK Competing interests: None declared
REVIEW RETURNED	11-Dec-2018

GENERAL COMMENTS	I have reviewed the paper from the perspective of the comments that I made on the first version that I saw. I think the responses to my comments are reasonable and appropriate given what could be added. I recognise that these are mostly limitations or thoughts for further research but this seems reasonable. Overall I think the paper is clear, well documented and will make an interesting contribution.
---